# The role of SPP1 in evaluating the prognosis, immune infiltration, and drug sensitivity of hepatocellular carcinoma

**Kai Cui**[1]☯, **Xia Li**[1]☯, **Yongrun Li**[2]☯, **Zhong Li**[3], **Du Wang**[1], **Xinhong Wang**[1], **Shuxin Qin**[1], **Junjie Li**[1], **Jiaye Long**[2]*

1 Department of Interventional Radiology, Inner Mongolia Forestry General Hospital, The Second Clinical Medical School of Inner Mongolia Minzu University, Yakeshi, Inner Mongolia, China, 2 Department of Interventional Radiology, Xinhui Hospital Affiliated to Southern Medical University, Xinhui District People's Hospital, Jiangmen, Guangdong, China, 3 Department of Medical Imaging, Inner Mongolia Forestry General Hospital, The Second Clinical Medical School of Inner Mongolia Minzu University, Yakeshi, Inner Mongolia, China

☯ These authors contributed equally to this work.
* 1145618270@qq.com

## Abstract

### Background

Secretory phosphoprotein 1 (SPP1) has been linked to tumor progression and immune regulation, but its prognostic value, impact on the tumor immune microenvironment (TIME), and drug sensitivity in HCC remain unclear.

### Methods

We performed a pan-cancer analysis using TIMER and validated SPP1 upregulation in six GEO datasets (GSE45436, GSE54236, GSE121248, GSE76427, GSE64041, and GSE60502) and HPA protein data. In TCGA-LIHC, we assessed overall survival (OS) and progression-free survival (PFS) using univariate/multivariate Cox analyses, ROC analysis, and a calibrated nomogram. We identified differentially expressed genes (DEGs) and performed GO/KEGG and GSEA analyses. Immune infiltration was estimated with CIBERSORT and TIMER, and relationships with immune checkpoints were explored. Drug sensitivity was predicted with pRRophetic using GDSC data. *In vitro*, SPP1 was knocked down or overexpressed in HCC cell lines to evaluate effects on proliferation, migration, invasion, and apoptosis via qRT-PCR, Western blot, CCK-8, colony formation, wound healing, Transwell invasion, and TUNEL assays.

### Results

SPP1 was significantly upregulated in HCC at mRNA and protein levels. High SPP1 predicted poorer OS and PFS and was associated with higher histological grade,

**Data availability statement:** the data are all contained within the manuscript and/or Supporting Information files.

**Funding:** This work was supported by the Science and Technology Program of the Joint Fund of Scientific Research for the Public Hospitals of Inner Mongolia Academy of Medical Sciences (Grant No. 2024GLLH0862). The funders had no role in study design, data collection and analysis, decision to publish, or preparation of the manuscript.

**Competing interests:** NO authors have competing interests.

**Abbreviations:** HCC: hepatocellular carcinoma, TACE: transarterial chemoembolization, OS: overall survival, SPP1: secretory phosphoprotein 1, TCGA: The Cancer Genome Atlas, GEO: Gene Expression Omnibus, HPA: Human Protein Atlas, PFS: progression-free survival, GO: Gene ontology, KEGG: Kyoto Encyclopedia of Genes and Genomes, GSEA: gene set enrichment analysis, GDSC: Genomics of Drug Sensitivity in Cancer, TME: tumor microenvironment, CSF-1: colony stimulating factor-1, LPS: lipopolysaccharide, IFN-γ: interferon-gamma, CTLs: cytotoxic T lymphocytes, EMT: epithelial-mesenchymal transition

advanced stage, and greater T stage. The nomogram showed good calibration and discrimination. DEGs and enrichment analyses implicated cytokine receptor interaction, fatty acid metabolism, and PI3K-Akt signaling; GSEA confirmed immune- and metabolism-related pathways. High SPP1 correlated with higher immune/ESTIMATE scores, increased M0/M2 macrophages and dendritic cells, reduced CD8＋T cells, and upregulation of multiple immune checkpoints. Drug-sensitivity predictions showed high-SPP1 tumors were more sensitive to several anti-cancer drugs (e.g., sorafenib), while resistance to others was suggested. Functionally, SPP1 knockdown inhibited, while overexpression promoted, proliferation, migration, and invasion; knockdown increased apoptosis.

## Conclusions

SPP1 acts as an oncogenic driver in HCC, associated with poor prognosis, an immunosuppressive TIME, and distinct drug-response patterns.

---

## Introduction

In 2020, hepatocellular carcinoma (HCC), a common malignant tumor, had a global burden of approximately 910,000 new cases and 830,000 deaths, ranking sixth and fourth, respectively, in terms of incidence and mortality worldwide [1]. Because early-stage HCC is often asymptomatic, about 70% of patients are diagnosed at an advanced stage and thereby lose the opportunity for curative surgical resection [2,3]. For unresectable HCC, transarterial chemoembolization (TACE), stereotactic radiotherapy, targeted therapy, and immunotherapy have become crucial treatment methods [4]. Despite these advances, the 5-year overall survival (OS) rate of HCC patients remains below 30% [5,6]. Hence, exploring effective molecular targets can aid in guiding clinical medication and surgical treatment, thereby improving patient prognosis.

Osteopontin, also termed secretory phosphoprotein 1 (SPP1), is defined as a secreted glycosylated phosphoprotein with a widespread presence in the extracellular matrix [7]. Initially, SPP1 was discovered in osteosarcoma cells [8]. Additionally, it is found in various tissues, including the kidneys, brain, breast, gastrointestinal tract, inner ear, and others [9–11]. SPP1 is present in both breast milk, semen, urine, and blood simultaneously [12,13]. SPP1 is a multifunctional protein that participates in a diverse range of processes, from biomineralization and bone formation to wound healing and anti-stone therapy [14,15]. However, overexpression of SPP1 can also promote inflammation, cardiovascular disease, insulin resistance, and tumor development [16–19]. Research has shown that SPP1 mediates chronic inflammatory infections and tumor development by regulating the host's immune response, especially in liver tissue [20]. SPP1 encodes a 294–amino-acid secreted phosphoprotein located on chromosome 4q13 [21]. SPP1 contains integrin-binding domains including RGD and SVVYGLR [22]. The RGD and SVVYGLR domains are the best characterized. SPP1 promotes tumor progression via interactions with CD44 and integrins.

The relationship between SPP1 and HCC has garnered considerable attention in recent years. Research has shown that overexpression of SPP1 has been detected in HCC patients [23]. Meanwhile, a recent meta-analysis comparing SPP1 with alpha fetoprotein in diagnosing HCC showed that SPP1 is more sensitive than AFP in diagnosing HCC [24]. Therefore, SPP1 has become a biomarker for diagnosing HCC.

Not only does SPP1 have an impact on tumor occurrence, but it is also linked to tumor progression, invasion, and metastasis [25]. According to an investigation by Zou et al. [26], SPP1 serves as an indicator to assess the prognosis and effectiveness of immunotherapy in patients with penile cancer. Park et al. [27] believe that preoperative SPP1 levels can predict muscle invasion and postoperative pathological staging of bladder urothelial carcinoma. SPP1 overexpression is considered a significant prognostic factor in lung cancer [28]. The research on how SPP1 progresses, invades, and metastasizes in HCC remains scarce. Additional clarification is necessary regarding the potential biological functions and prognostic significance of SPP1 in HCC.

This study aims to comprehensively elucidate the role of SPP1 in HCC progression and therapy. To achieve this, we performed a multi-faceted study. First, we assessed the mRNA and protein expression levels of SPP1 in HCC using publicly available databases, including The Cancer Genome Atlas (TCGA), Gene Expression Omnibus (GEO), and the Human Protein Atlas (HPA). Subsequently, the prognostic value of SPP1 in HCC was further validated with TCGA survival data. We also investigated the association between SPP1 expression and immune cell infiltration within the tumor micro-environment. In addition, the potential of SPP1 as a biomarker for predicting drug sensitivity in HCC was systematically evaluated. To experimentally verify its functional role, SPP1 was knocked down or overexpressed in HCC cell lines, and its effects on proliferation, migration, invasion, and apoptosis were examined *in vitro*.

## Materials and methods

### Differential gene expression analysis

To assess the mRNA expression profile of SPP1 across various cancers, a pan-cancer analysis was conducted using the TIMER database, which covers 33 tumor types. Subsequently, six HCC datasets (GSE45436, GSE54236, GSE121248, GSE76427, GSE64041, and GSE60502) were obtained from the GEO database to confirm elevated SPP1 mRNA levels in HCC patients. Differential expression of SPP1 between HCC and non-HCC tissues was analyzed and visualized using the ggplot2 package in R. Furthermore, protein expression data for SPP1 in HCC and corresponding non-tumor tissues were retrieved from the HPA database. The data used in this study were all from public databases, and the cell lines used were commercially purchased. Therefore, this study did not require specific ethical approval.

### Survival and clinicopathological analysis in HCC patients

To evaluate the impact of SPP1 expression on OS and progression-free survival (PFS) in HCC patients, we used "survival" packages to analyze HCC cohort samples downloaded from TCGA. We represented them in the form of Kaplan-Meier curves. Univariate and multivariate Cox regression analyses were performed using the "survival" package to explore the correlation between SPP1 expression and clinicopathological factors in HCC. Furthermore, we assessed the prognostic predictive value of SPP1 for HCC patients using receiver operating characteristic (ROC) curves. Subsequently, a nomogram was constructed based on the aforementioned clinicopathological factors, and a calibration curve was utilized to assess the consistency of the model.

### Functional enrichment analysis

374 individuals with LIHC in TCGA were categorized into high- and low-expression groups based on the median level of SPP1 mRNA. Two groups of differentially expressed genes (DEGs) were screened using the threshold criteria of |logFC| > 1 and FDR < .05. Using the "pheatmap" package, the results of the aforementioned DEGs were shown as

heatmaps. "cluster Profiler" and "org. Hs. e.g., db" packages were used for Gene ontology (GO) and Kyoto Encyclopedia of Genes and Genomes (KEGG) enrichment analysis, presented in the form of bar charts. Additionally, we conducted gene set enrichment analysis (GSEA) to identify the potential regulatory pathways of SPP1.

### Correlation analysis between SPP1 and immune cell infiltration

To compare the infiltration levels of 22 distinct kinds of immune cells that infiltrate tumors between the SPP1 low-expression group and the SPP1 high-expression group, we employed the "CIBERSORT" package. Use TIMER to analyze the correlation between SPP1 expression and tumor immune-infiltrating cells. In addition, we also evaluated the relationship between SPP1 expression and immune/matrix/estimated scores. Ultimately, we assessed the association between SPP1 and genes relevant to immunological checkpoints. The above analysis was implemented using "CIBER-SORT", "reshape2", "ggpubr", "vioplot", "ggExtra", and "corrplot" packages.

### Drug discovery and prediction

The IC50 was utilized to assess the efficacy of drugs for treating HCC. The drugs used for evaluating cancer treatment were derived from the Genomics of Drug Sensitivity in Cancer (GDSC) database. The above analysis was carried out using the "pRRopetic" package.

### Cell culture

This study utilized multiple cell lines, including the human normal HCC line LO2, and the human HCC cell lines HepG2, Hep3B, Huh-7, Bel-7402, and SNU-387. All cells were cultured under standard conditions (37°C, 5% $CO_2$) using DMEM medium supplemented with 10% fetal bovine serum. To identify a representative HCC cell line for functional assays, the basal expression levels of SPP1 were first evaluated in a panel of human HCC cell lines and the normal human liver cell line. Total RNA was extracted from these cell lines using TRIzol reagent (Invitrogen) according to the manufacturer's protocol. cDNA was synthesized from 1–2 μg of total RNA using a PrimeScript RT reagent kit (Takara). Quantitative real-time PCR (qRT-PCR) was performed with SYBR Green PCR master mix (Applied Biosystems) on a QuantStudio 3 Real-Time PCR System.

### Cell transfection

SPP1 expression was transiently manipulated in HCC cell lines using siRNA-mediated knockdown and plasmid-mediated overexpression. For knockdown experiments, cells were transfected with SPP1-specific siRNAs (si-SPP1–1, si-SPP1–2, and si-SPP1–3) or a negative control siRNA. For overexpression, cells were transfected with an SPP1 expression plasmid (OE-SPP1) or an empty vector control. Transfections were performed usingLipofectamine 3000 according to the manufacturer's instructions.

### qRT-PCR

Total RNA was extracted using TRIzol and analyzed using a NanoDrop. cDNA was synthesized from 1 to 2 μg of RNA. qRT-PCR was conducted using SYBR Green, with the following protocol: 95°C for 5 minutes, followed by 40 amplification cycles (95°C for 10 seconds and 60°C for 30 seconds), and then melt-curve analysis. SPP1 transcript levels were determined relative to GAPDH via the 2^-ΔΔCt method. Primer sequences are available in S1 Table.

### Western Blot

Cell lysates prepared with RIPA buffer were quantified by the BCA method. Equal amounts of protein were subjected to SDS-PAGE and transferred to PVDF membranes. Membranes were blocked with 5% skim milk for 1 hour at room

temperature, then probed overnight at 4°C with anti-SPP1 antibody (1:1000). After TBST washes, membranes were incubated with HRP-linked secondary antibody (1:5000) for 1 hour at room temperature. Detection was performed with an ECL substrate. GAPDH (1:1000) acted as an internal control, and semi-quantification was conducted using ImageJ.

## CCK-8 assay

Logarithmically growing HepG2 cells were plated at an appropriate density in 96-well plates. After 24 hours to allow for cell adhesion, the medium was exchanged for fresh medium containing the various treatment agents. From day 0 to day 4, CCK-8 reagent was added to the cultures daily at a fixed time point. Following a 2–4 hour incubation, the absorbance (OD) at 450 nm was measured using a microplate reader. A cell viability growth curve was then generated from the obtained data.

## Colony formation assay

500 logarithmically growing HepG2 cells were seeded per well of a 6-well plate. Cells were cultured under standard conditions (37°C, 5% $CO_2$) for 2–3 weeks, with medium replaced every 3–4 days. After incubation, colonies were washed twice with PBS, fixed with 4% paraformaldehyde for 15 minutes, and stained with 0.1% crystal violet for 30 minutes. Following a gentle tap-water rinse to remove excess dye, plates were air-dried. Colonies were counted under a microscope, and the colony formation rate was calculated.

## Wound healing assay

To assess cell migration using a scratch assay, HepG2 cells were cultured in 6-well plates to near confluence. A sterile pipette tip was used to create a linear wound in the monolayer. After removing detached cells with three PBS washes, the cells were incubated in serum-free medium. Microscopic images of the wound were captured immediately and after 24 hours at identical positions. The relative wound area closure over 24 hours was then quantified by analyzing the images with ImageJ software.

## Transwell invasion assay

Log-phase HepG2 cells were collected and resuspended at $1 \times 10^5$ cells/mL in serum-free medium. The upper surface of Matrigel-coated Transwell membranes was coated with 50–100 µL of Matrigel (1:8) and allowed to gel at 37°C for 1–3 h. Add 200 µL of the cell suspension to the upper chamber and 600 µL of complete medium with 10% FBS to the lower chamber. Incubate at 37°C, 5% CO2 for 24 h. Remove non-migrated cells from the upper surface. Fix migrated cells on the lower surface with 4% paraformaldehyde for 30 minutes and then stain with 0.1% crystal violet for 20 minutes.

## TUNEL assay

Apoptosis detection was performed using the TUNEL method after fixation (4% PFA, 20 min) and permeabilization (0.1% Triton X-100, 2 min). TUNEL labeling was performed at 37°C in the dark for 1 h, followed by DAPI counterstaining (5 min).

## Statistical analyses

Statistical analyses were performed using R (version 4.1.0) and GraphPad Prism (version 10.0). Data from *in vitro* experiments are presented as mean ± standard deviation (SD) from at least three independent experiments. Differences between the two groups were assessed using a two-tailed Student's t-test or a Wilcoxon rank-sum test, as appropriate. Comparisons among multiple groups were performed using one-way analysis of variance (ANOVA) followed by post hoc tests. Categorical variables were compared using the chi-square ($\chi^2$) test. P-values < .05 were considered statistically significant.

## Results

### Upregulation of SPP1 in HCC tissues

TIMER database showed that the mRNA expression levels in 20 tumor tissues, including HCC tissue, outpaced those in non-tumor tissues in 33 human tumors (Fig 1A). The difference in SPP1 expression between the HCC group and the non-HCC group in the TCGA-LIHC cohort was shown in Figs 1B and 1C. Furthermore, data from six HCC datasets (GSE45436, GSE54236, GSE121248, GSE76427, GSE64041, and GSE60502) obtained from the GEO database consistently confirmed higher SPP1 expression in HCC relative to non-HCC tissues (Figs 1D-I). According to the HPA database, HCC tissues also exhibited increased SPP1 protein expression compared to non-HCC tissues (Fig 1J).

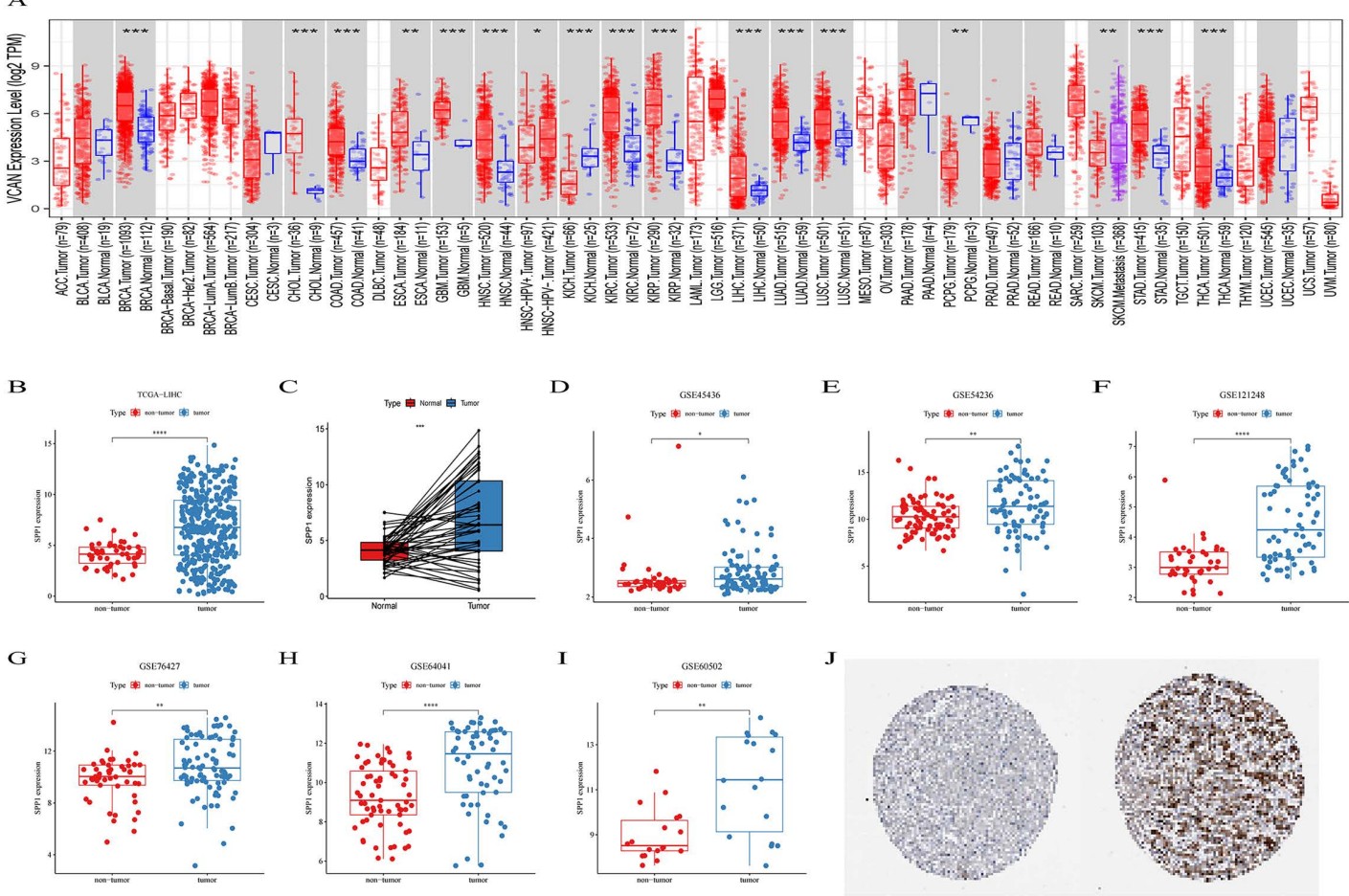

**Fig 1. The expression of SPP1 in normal and liver cancer tissues. (A)** SPP1 gene expression in 33 tumor tissues in TIMER database. **(B)** Compared with normal samples, SPP1 expression is higher in HCC samples. **(C)** SPP1 was significantly elevated in 374 HCC samples compared to the corresponding adjacent liver tissue. **(D-I)** In databases GSE45436, GSE54236, GSE121248, GSE76427, GSE64041, and GSE60502, SPP1 expression is higher in HCC samples compared to normal samples. **(J)** In the HPA database, the expression of SPP1 protein is elevated in HCC tissues (right circle) compared to non HCC tissues (left circle).

## Association between SPP1 expression and prognosis and clinicopathological features

We analyzed SPP1 expression using clinical data from the TCGA-LIHC cohort. Survival analysis revealed significant differences in both OS and PFS between patients with high and low SPP1 expression (Figs 2A and 2B). SPP1 expression was significantly elevated in tumors of higher histological grades (G2, G3, and G4) compared to grade G1 (Fig 2C). Similarly, expression was higher in advanced pathological stages (II and III) relative to stage I (Fig 2D) and in T2 and T4 stages compared to T1 (Fig 2E). No significant associations were found between SPP1 expression and patient gender, age, or N and M stages. Furthermore, SPP1 demonstrated considerable predictive power for OS in HCC patients, as indicated by the area under the receiver operating characteristic curve (AUC) (Fig 2F). Significant differences in SPP1 expression were confirmed across histological grades, pathological stages, and T stages (Fig 2G). Univariate and multivariate Cox regression analyses identified SPP1 expression and pathological stage as independent risk factors affecting HCC patient outcomes (Figs 2H and 2I). Based on these variables, a nomogram was constructed to predict OS probability (Fig 2J). The calibration curve indicated strong concordance between the nomogram's predictions and actual observed survival (Fig 2K).

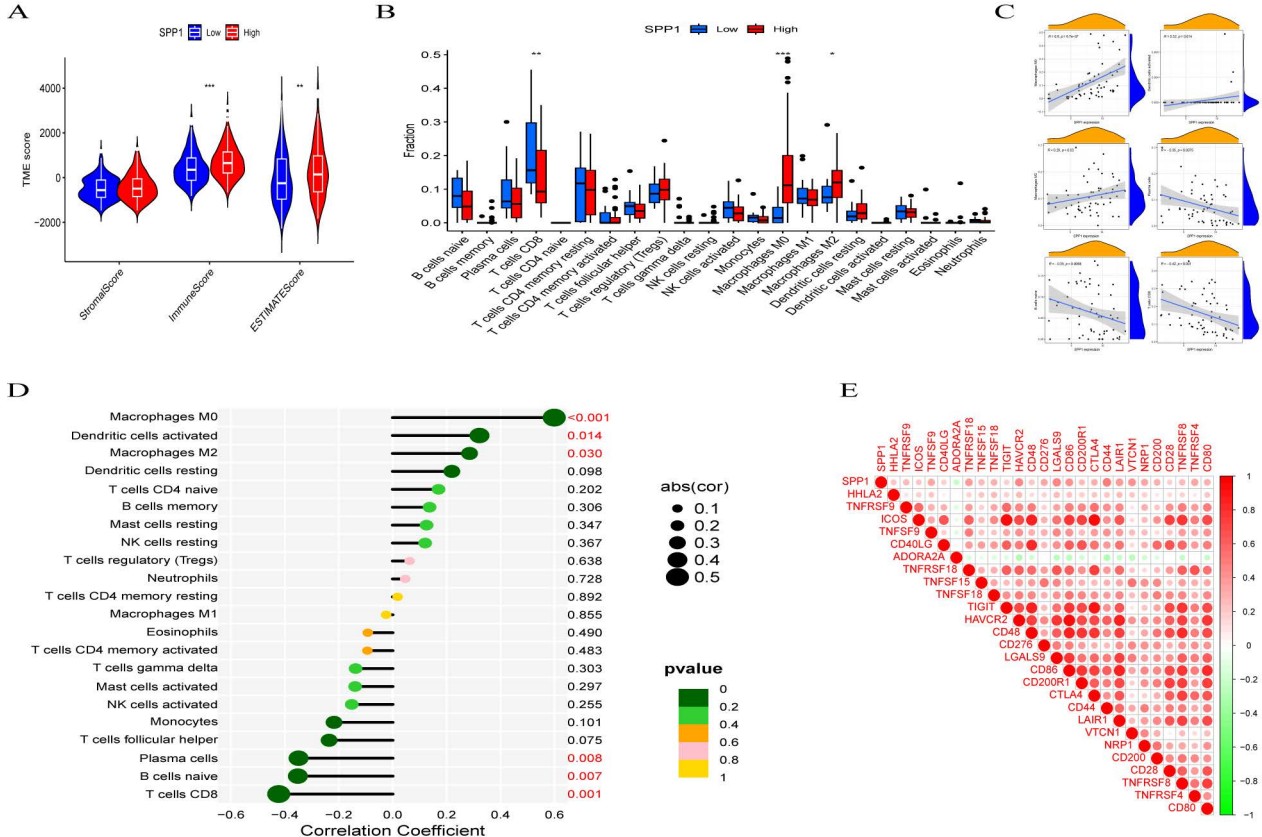

**Fig 2. Clinical characteristics of SPP1 expression in HCC patients. (A)** The relationship between SPP1 differences and overall survival in HCC patients. **(B)** The relationship between SPP1 differences and progression free survival in HCC patients. **(C)** The relationship between SPP1 differences and histological grading of HCC patients. **(D)** The relationship between SPP1 differences and histological grading of HCC patients in pathological stages. **(E)** The relationship between SPP1 differences and T grading in HCC patients. **(F)** ROC curve for predicting 1/3/5-year survival rate using SPP1. **(G)** Heat map showing the correlation between SPP1 expression differences and clinical features. Univariate **(H)** and multivariate **(I)** Cox regression analysis of SPP1 with clinical features and survival in HCC patients. **(J)** Nomogram for predicting survival of HCC patients based on SPP1 expression. **(K)** Calibration curve for predicting 1/3/5-year survival rate of patients using Nomogram.

## Potential pathways of SPP1 regulation in HCC

The median level of SPP1 was used to divide the two groups of HCC patients into high- and low-SPP1 expression groups. After adjusting, we discovered 2,536 DEGs, which consisted of 2,267 upregulated genes and 269 downregulated genes. Among them, the 50 most significant gene-related heatmaps were shown in Fig 3A. GO analysis revealed that these

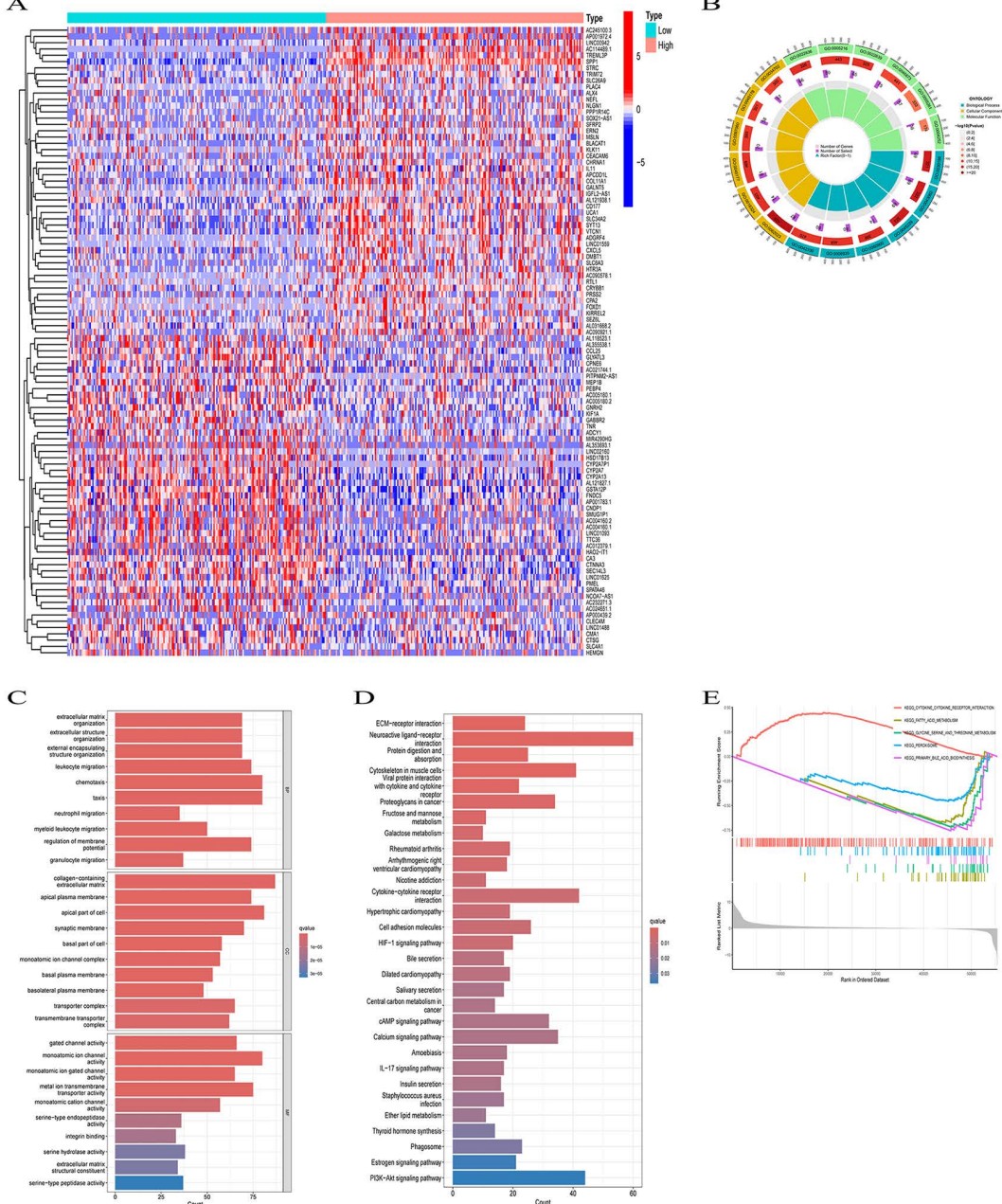

**Fig 3. Functional enrichment analysis of HCC. (A)** Heat map of the 50 most important related genes in the SPP1 high and low expression groups. **(B)** Circle graph of GO enrichment analysis. **(C)** GO annotation of differentially expressed genes in HCC. **(D)** Bar plot of KEGG enrichment analysisr esults. **(E)** GSEA analysis of SPP1 overexpression.

DEGs were primarily enriched in biological processes such as taxis, chemotaxis, leukocyte migration, and regulation of the membrane (Figs 3B and 3C). In the cellular component, DEGs were mainly concentrated in the collagen-containing extracellular matrix, the apical part of the cell, the apical plasma membrane, and the synaptic membrane, among others (Figs 3B and 3C). The KEGG enrichment pathway primarily included neuroactive ligand-receptor interaction, cytoskeleton in muscle cells, cytokine-cytokine receptor interaction, and the PI3K-Akt signaling pathway, among others (Fig 3D). GSEA analysis revealed that the high expression of SPP1 was associated with cytokine receptor interaction, fatty acid metabolism, metabolism of glycine, serine, and threonine, peroxisome function, and primary bile acid biosynthesis (Fig 3E).

## Correlation of SPP1 Expression with Tumor Immune Infiltration

There was a statistically significant difference in immune scores and ESTIMATE scores between the low SPP1 group and the high SPP1 group (Fig 4A). Immune cell infiltration analysis further indicated that elevated SPP1 expression was associated with a higher proportion of M0 macrophages and a lower proportion of CD8+T cells (Fig 4B). Additionally, SPP1 expression showed positive correlations with M0 macrophages (r=0.60, P<.001), M2 macrophages (r=0.29, P=.03), and

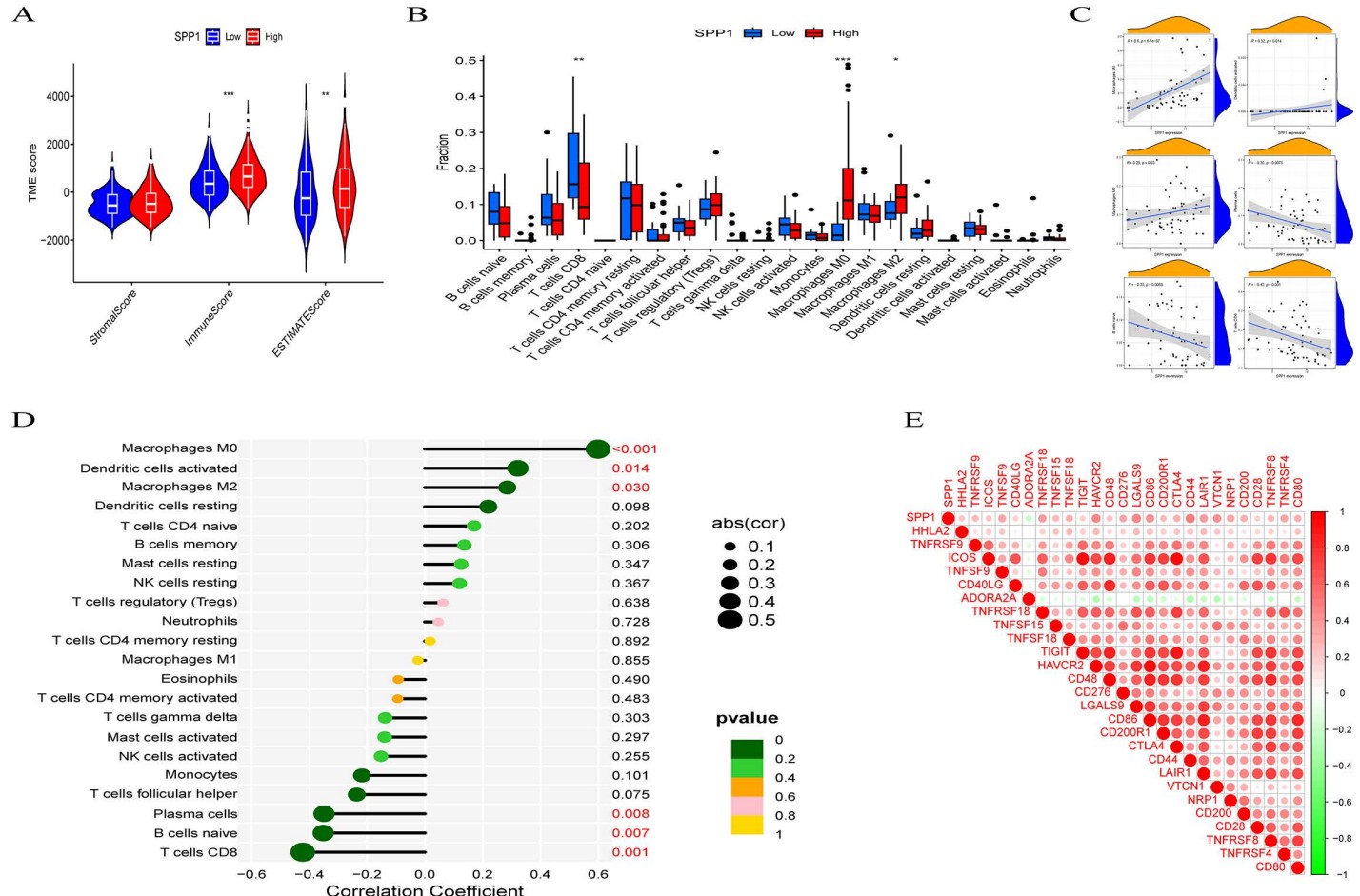

**Fig 4. Correlation between SPP1 and immune infiltration in HCC. (A)** High expression of SPP1 is associated with immune score and estimated score. **(B)** The relationship between SPP1 expression and immune/stromal/assessment scores in HCC. **(C)** The impact of differential expression of SPP1 on immune cell infiltration. **(D)** The correlation between SPP1 expression and immune cell infiltration. **(E)** The correlation between SPP1 and immune checkpoint-related genes. (*P<.05; **P<.01; ***P<.001).

activated dendritic cells (r = 0.32, P = .014) (Fig 4C), while demonstrating negative correlations with CD8$^+$T cells (r = −0.42, P = .001), naïve B cells (r = −0.35, P = .007), and plasma cells (r = −0.35, P = .008) (Fig 4D). Moreover, SPP1 expression was notably correlated with 26 immune checkpoint-related genes (P < .001), showing positive associations with 25 of these genes and a negative correlation with ADORA2A (Fig 4E).

## The relationship between SPP1 and drug sensitivity in HCC

We assessed the drug sensitivity of various SPP1 expression levels using IC50. The IC50 values for 17-AAG, A-770041, AG-014699, BI-2536, Bortezomab, CGP-60474, CGP-082996, Dasatinib, FR-180204, GW843682X, imatinib, JW-7-52-1, KIN001−102, LY317615, mitomycin C, MS-275, paclitaxel, pyrimethamine, sorafenib, S-Trityl-L-cysteine, sunitinib, TAE684, THZ-2–49, VX-680, and WZ-1–84 were lower in patients with high expression of SPP1 expression (Fig 5, S2 Table). Transcriptomic drug sensitivity analysis predicted that the high-SPP1 subgroup may exhibit increased sensitivity to these medications. Higher IC50 values were observed for (5Z) −7-Oxozeaenol, BX-795, camptothecin, CEP-701, CI-1040, CX-5461, EHT 1864, eleslimol, FH535, FTI-277, GDC0449, GSK429286A, JNJ-26854165, KIN001−135, KU-55933, methodrexate, NSC −207895, NU-7441, OSI-027, PI-103, piperlongumine, QL-X-138, QL-XI-92, QL-XII-47, SN-38, SNX-2112, temsirolimus, THZ-2-102-1, tramenib, TW 37, Vorinostat, VX-702, XMD13−2, YM155, and ZSTK474 (Fig 6, S2 Table). This suggested that individuals with HCC who expressed high levels of SPP1 were not candidates for therapy with the previously stated medications.

## Effects of SPP1 on proliferation, migration, invasion, and apoptosis of HCC cells

To validate the findings from the in silico analysis, we modulated SPP1 expression in HCC cell lines using transient transfection. To validate bioinformatics findings, SPP1 expression was examined in HCC cell lines. qRT-PCR revealed significantly

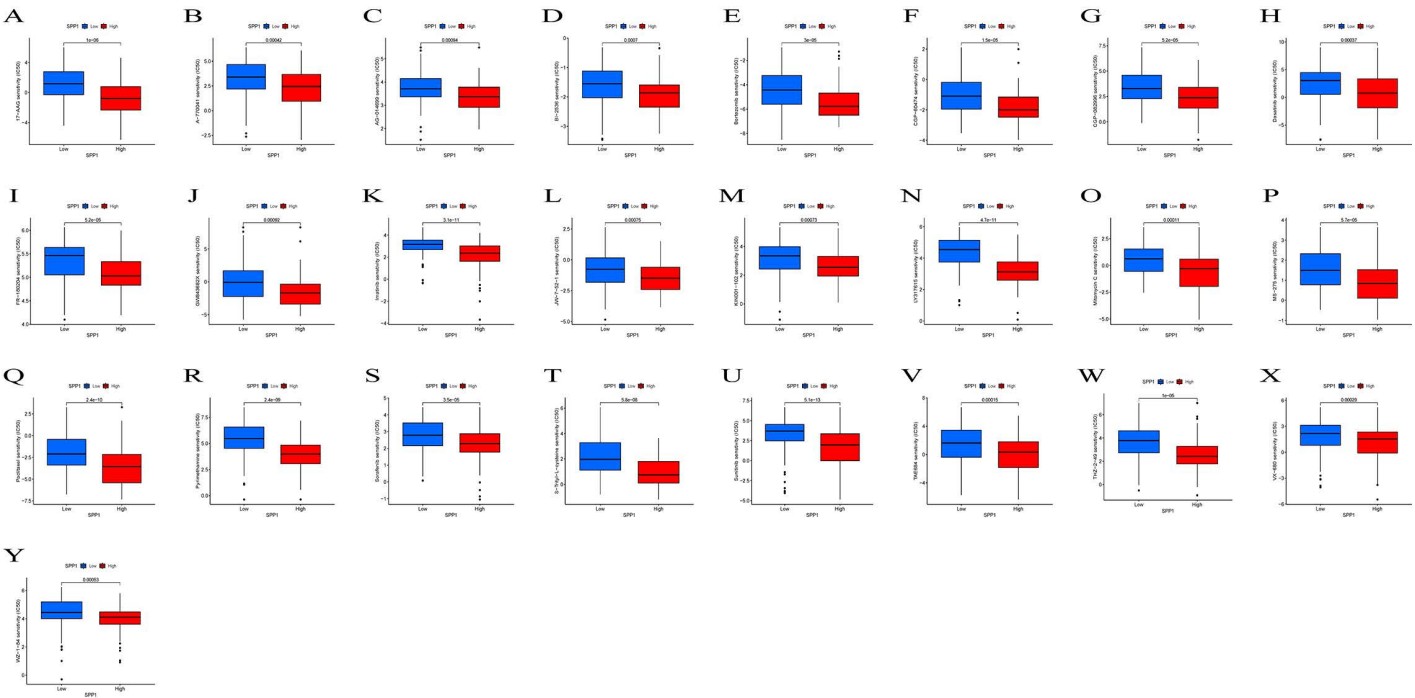

**Fig 5. Drugs with low IC50 values in HCC patients with high SPP1 expression.** High-SPP1 subgroup may exhibit increased sensitivity to these medications.

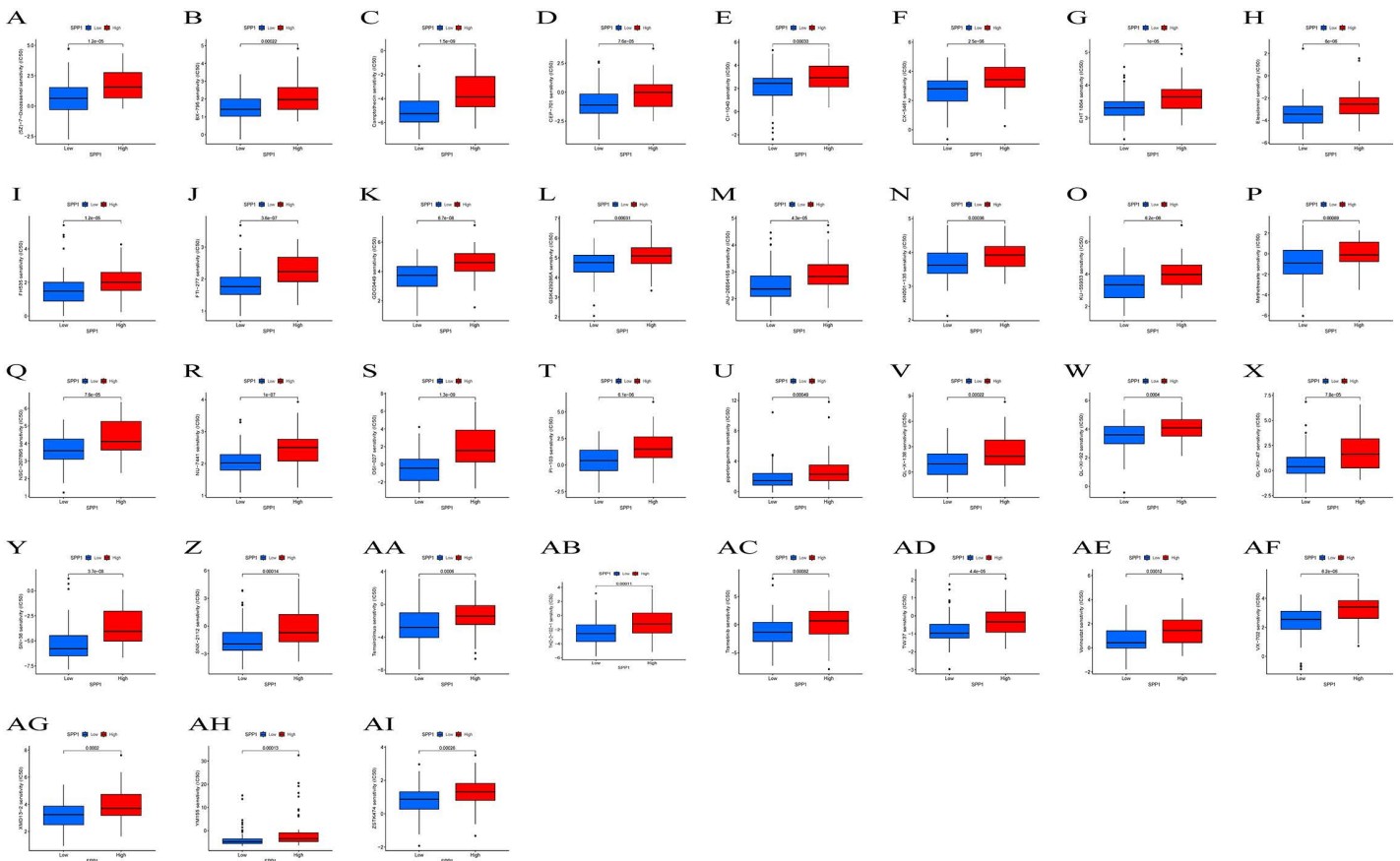

**Fie 6. Drugs with higher IC50 values in HCC patients with high SPP1 expression.** High-SPP1 subgroup may exhibit decreased sensitivity to these medications.

elevated SPP1 mRNA levels in HepG2, Huh-7, and Bel-7402 cells compared to normal hepatocytes LO2 (Fig 7A). This upregulation was further confirmed at the protein level in HepG2, Hep3B, and Bel-7402 cells by Western blot (Figs 7B and C). To investigate the function of SPP1, HepG2 cells were transfected with SPP1-specific siRNA (si-SPP1) or an overexpression plasmid (OE-SPP1). QRT-PCR and Western blot verified successful knockdown and overexpression compared to respective controls (Figs 7D-7F).

Functional assays demonstrated that SPP1 knockdown significantly suppressed HepG2 cell proliferation, whereas overexpression enhanced it, as assessed by CCK-8 (Fig 8A). Colony formation assays indicated reduced colony number and size upon SPP1 silencing, whereas SPP1 overexpression promoted clonogenicity (Fig 8B). In wound healing experiments, SPP1-overexpressing cells displayed the highest migration rate at 24h, whereas knockdown cells migrated the least (Figs 8C and 8D). Consistent with this, Transwell invasion assays demonstrated that SPP1 overexpression increased cell penetration, while knockdown impaired invasiveness (Figs 8E and 8F). Lastly, TUNEL assays revealed that SPP1 overexpression decreased apoptosis, whereas knockdown elevated the apoptotic rate in HepG2 cells (Figs 8G and 8H).

## Disscussion

In this work, we employed bioinformatics techniques to assess the role of SPP1 in the expression, prognosis, and therapeutic sensitivity of HCC patients, and validated our findings through qRT-PCR, Western blotting, and functional assays

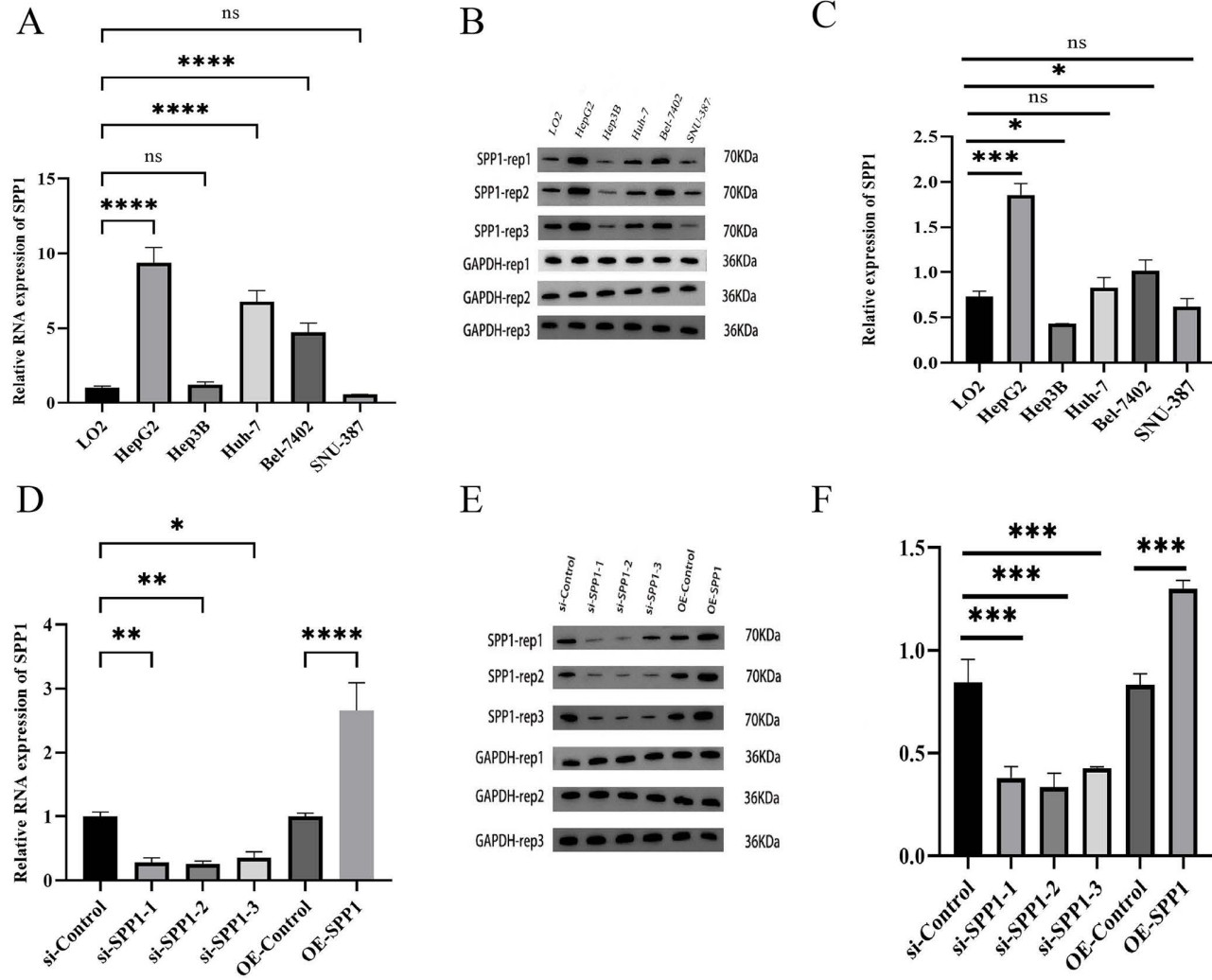

**Fig 7. Expression and inhibition of SPP1 in HCC cell lines. (A)** Relative mRNA expression of SPP1 in various cell lines. **(B–C)** Western blot analysis of SPP1 protein expression in different cell lines. **(D)** Relative mRNA expression of SPP1 after knockdown with different siRNAs. **(E–F)** Western blot analysis of SPP1 protein expression after siRNA knockdown. (*P<.05; **P<.01; ***P<.001; ****P<.0001).

including CCK-8, colony formation, wound healing, Transwell invasion, and TUNEL assays. According to research, SPP1 is highly expressed in pan-cancer analysis, including HCC. We validated SPP1 upregulation using six independent GEO cohorts. Clinically, high SPP1 correlates with poorer OS and PFS. Multivariate analysis indicates SPP1 is an independent prognostic factor and associates with higher histologic grade and T stage.

Wang et al. [29] reported that SPP1 activates immune cells within the tumor microenvironment (TME), promoting the progression of esophageal squamous cell carcinoma. Therefore, we investigated the relationship between SPP1 expression and immune-related features in HCC. In our analyses, SPP1 expression was associated with transcriptional signatures related to neutrophils, macrophages, myeloid dendritic cells, and B cells. Furthermore, six distinct types of invading immune cells are significantly associated with SPP1. M0 macrophages and CD8$^+$T cells had the strongest positive connection (r=0.6) and the most significant negative correlation (r=0.4), respectively, among these. Macrophages, derived from monocytes, are key effectors of innate and adaptive immunity and exhibit phenotypic plasticity in tumors;

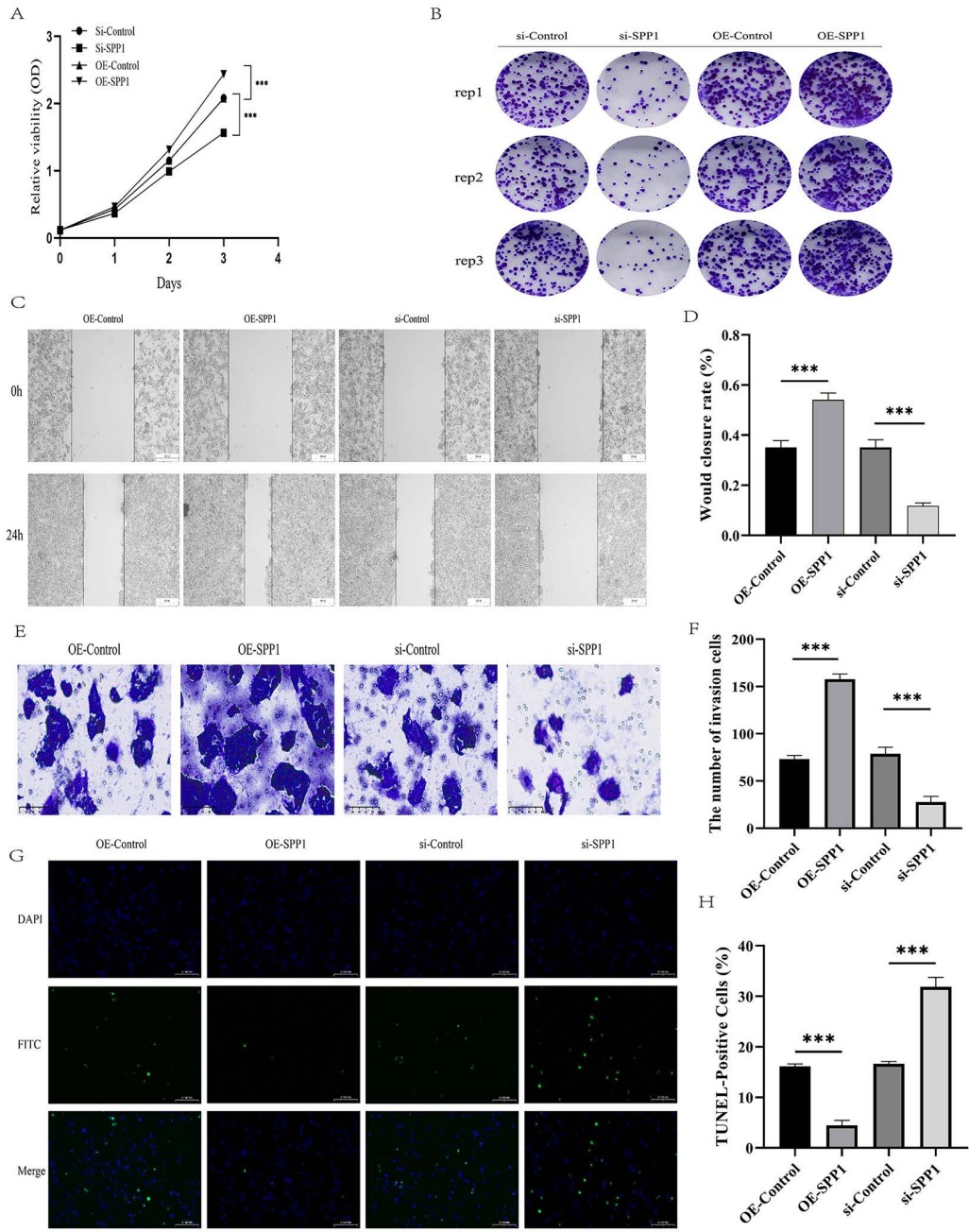

**Fig 8. Effects of SPP1 on the proliferation, clonogenicity, migration, invasion, and apoptosis of HCC cells. (A)** Cell proliferation of HepG2 cells after SPP1 knockdown or overexpression was assessed by the CCK-8 assay. **(B)** Colony formation ability of HepG2 cells was evaluated after SPP1 knockdown or overexpression. **(C-D)** Cell migration was assessed by wound healing assay in HepG2 cells with SPP1 knockdown or overexpression. **(E-F)** Cell invasion was evaluated by Transwell assay in HepG2 cells with SPP1 knockdown or overexpression. **(G-H)** Apoptosis of HepG2 cells was examined by TUNEL staining after SPP1 knockdown or overexpression. (*P < .05; **P < .01; ***P < .001).

different macrophage phenotypes can exert distinct and sometimes opposing effects on tumor biology [30,31]. *In vitro*, colony-stimulating factor-1 (CSF-1) induces M0 macrophage activation. M0 macrophages are capable of phagocytos-ing pathogens, but their full function requires polarization [32]. For example, exposure to lipopolysaccharide (LPS) and

interferon-gamma (IFN-gamma) polarizes M0 macrophages toward an M1 phenotype, which exerts anti-tumor effects through cytotoxicity and antibody-dependent cellular cytotoxicity [33,34]. In contrast, cytokines such as IL-4, IL-13, IL-10, IL-33, and TGF-β drive polarization toward the M2 phenotype, which promotes immunosuppression, angiogenesis, and tumor growth [ 35–37]. CD8+T cells are a type of cell that expresses CD8 molecules on the surface of T lymphocytes and differentiate into functional cytotoxic T lymphocytes (CTLs) by binding to antigens [38,39]. CD8+T cells kill tumor cells by secreting proteases, perforin, and IL [40,41]. In addition, CD8+T cells induce tumor cell apoptosis by binding to Fas-L receptors [42]. According to Upadhyay et al. [43], SIGLEC10+macrophages suppress the activity of CD8+T cells, thereby aiding in the development of gastric cancer. SPP1 showed a negative correlation with CD8+T cells and a positive correlation with M0 macrophages in this investigation. Thus, research into the possible mechanism of SPP1 differential expression between M0 macrophages and CD8+T cells is required.

In addition to promoting proliferation, migration, and invasion, SPP1 may also contribute to hepatocellular carcinoma progression by facilitating epithelial-mesenchymal transition (EMT) and metastatic dissemination. EMT is a crucial biological process in tumor progression, during which epithelial cells lose cell-cell adhesion and acquire mesenchymal characteristics, thereby enhancing motility, invasiveness, and resistance to apoptosis. Previous studies have shown that SPP1 is closely associated with EMT-related phenotypes in multiple malignancies and may regulate the expression of canonical EMT markers, including E-cadherin, N-cadherin, and Vimentin, through pathways such as PI3K/AKT, NF-κB, and TGF-β signaling [44,45]. In HCC, elevated SPP1 expression has been reported to correlate with more aggressive clinicopathological features and poor prognosis, suggesting a potential role in metastatic progression [46,47]. In the present study, SPP1 overexpression significantly enhanced the migratory and invasive abilities of HepG2 cells, whereas SPP1 knockdown produced the opposite effect, which is consistent with a pro-EMT and pro-metastatic role of SPP1. Although EMT markers and in vivo metastatic behavior were not directly evaluated in our study, these findings, together with previous reports, suggest that SPP1 may promote HCC aggressiveness at least partly by participating in EMT-related phenotypic plasticity and metastatic potential. Further studies are needed to determine whether SPP1 directly regulates EMT marker expression and metastatic colonization in HCC.

Numerous anti-cancer medications are extremely sensitive to HCC patients who have high expression levels of SPP1, according to drug sensitivity research. For instance, sorafenib is now the first-line treatment for advanced HCC, but only 30% of patients eventually see benefit from it, and most patients develop resistance to it within six months of starting to take it [48]. The potential mechanisms of sorafenib resistance can be explained from the perspectives of epigenetics, transport processes, cell death, and TME [49]. In our data, high SPP1 expression correlated with greater sensitivity to sorafenib, suggesting SPP1 could be a potential therapeutic target in HCC. Imatinib, as a tyrosine kinase inhibitor, has a good therapeutic effect on C-Kit mutated gastrointestinal stromal tumors [50]. Therefore, we speculate that C-Kit mutations in HCC are associated with SPP1 expression. Mitomycin and paclitaxel, respectively, are broad-spectrum anti-tumor drugs for antibiotic chemotherapy and anti microtubule chemotherapy. According to the NCCN guidelines, oxaliplatin, fluorouracil, and folic acid (FOLFOX) regimens have become the first-line chemotherapy options for HCC [51,52]. However, due to HCC being a naturally insensitive tumor to chemotherapy drugs, even with first-line chemotherapy regimens, most patients will still have a poor prognosis. At the same time, instead of developing new chemotherapy drugs, selecting appropriate chemotherapy drugs based on targets is more cost-effective. Our findings on mitomycin C and paclitaxel sensitivity provide a rationale for considering these drugs in systemic or locoregional chemotherapy for high-SPP1 HCC. On the contrary, Camptothecin, Methotrexate, and Temsirolimus have higher IC50 values in patients with high SPP1 expression, indicating that these drugs are not suitable for patients with high SPP1 expression. These drugs are also common anti-tumor drugs. Conversely, high SPP1 expression was associated with resistance to camptothecin (a topoisomerase-I inhibitor), methotrexate (a dihydrofolate reductase inhibitor), and temsirolimus (an mTOR inhibitor) [53–55]. This suggests that SPP1-mediated resistance may interfere with the mechanisms of these drugs, such as DNA replication inhibition, folate metabolism, or PI3K/AKT/mTOR signaling. Further studies are needed to elucidate how SPP1 expression drives these resistance pathways in HCC.

To further investigate the function of SPP1 in HCC, we performed a series of *in vitro* studies. Using siRNA-mediated knockdown, we found that suppression of SPP1 expression significantly reduced the proliferative, migratory, and invasive capacities of HCC cells. These results demonstrate that SPP1 actively contributes to HCC aggressiveness by enhancing both tumor growth and invasive potential. Consistent across multiple experimental approaches, our findings suggest that targeting SPP1 could be a promising therapeutic strategy to inhibit HCC progression and metastasis, with potential implications for improving patient outcomes.

In this study, we provide a multi-layered characterization of SPP1 in HCC. Across TCGA, six GEO datasets, and HPA protein data, SPP1 was consistently upregulated at the mRNA and protein levels. High SPP1 expression was associated with poorer OS and PFS. *In vitro*, SPP1 knockdown reduced HCC cell proliferation, migration, and invasion and increased apoptosis, while SPP1 overexpression had opposite effects.

In summary, our study supports the potential of SPP1 as a prognostic biomarker and therapeutic target in HCC. This article, however, presents several limitations. Firstly, our experimental approach was limited to a basic validation of the involvement of SPP1 in hepatocellular carcinoma (HCC). Further investigation is necessary to elucidate the role of SPP1 in the initiation and progression of HCC. Secondly, while our study identified a significant correlation between SPP1 and tumor immune infiltration, the precise mechanisms underlying this relationship remain unclear and warrant additional research for confirmation. Thirdly, *in vitro* experiments only utilize the HepG2 cell line. The effects of SPP1 on cell proliferation, migration, and invasion should be validated in more HCC cell lines and animal models. Additionally, while we analyzed tumor immune microenvironment and immune cell infiltration related to SPP1 expression using computational estimations, we did not experimentally investigate the interactions between stromal cells, resident liver cells, and infiltrating immune cells. Future research should address these aspects to enhance the interpretability and robustness of our findings. Finally, drug sensitivity prediction is based on cell line data from the GDSC database, which may differ from the actual therapeutic effect in clinical patients. It is necessary to validate these results in combination with clinical samples.

## Conclusions

In summary, this study demonstrates that SPP1 is highly expressed in HCC and correlates with advanced disease, unfavorable prognosis, an immunosuppressive tumor microenvironment, and distinct drug sensitivity patterns. Functional experiments reveal that SPP1 promotes HCC cell proliferation, migration, invasion, and survival. These results suggest that SPP1 could serve as both a prognostic biomarker and a promising therapeutic target, offering a potential direction for personalized treatment in HCC. Further mechanistic investigations and clinical validation are needed to advance SPP1-directed strategies toward clinical application.

## Supporting information

**S1 Table. qRT-PCR Protocol and Primer Sequences.**
(DOCX)

**S2 Table. Comparison of IC50 values of candidate drugs between SPP1-low and SPP1-high HCC groups.**
(DOCX)

**S1 Fig. Raw images.**
(PDF)

## Author contributions

**Conceptualization:** Kai Cui, Xia Li, Yongrun Li.

**Methodology:** Zhong Li, Du Wang.

**Software:** Xinhong Wang, Shuxin Qin.

**Visualization:** Junjie Li.

**Writing – original draft:** Jiaye Long.

**Writing – review & editing:** Jiaye Long.

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
