## [Decision Letter · Decision Letter 0]

17 Mar 2026

Dear Dr. Long,

Thank you for submitting your manuscript to PLOS ONE. After careful consideration, we feel that it has merit but does not fully meet PLOS ONE’s publication criteria as it currently stands. Therefore, we invite you to submit a revised version of the manuscript that addresses the points raised during the review process.

We look forward to receiving your revised manuscript.

Kind regards,

Vinay Kumar, Ph.D.

Academic Editor

PLOS One

**Journal Requirements:**

https://journals.plos.org/plosone/s/file?id=wjVg/PLOSOne_formatting_sample_main_body.pdf andandandand

“the Science and Technology Program of the Joint Fund of Scientific Research for the Public Hospitals of Inner Mongolia Academy of Medical Sciences (Grant No. 2024GLLH0862).”

6. Please include captions for your Supporting Information files at the end of your manuscript, and update any in-text citations to match accordingly. Please see our Supporting Information guidelines for more information: http://journals.plos.org/plosone/s/supporting-information....

Reviewers' comments:

Reviewer's Responses to Questions

**Comments to the Author**

1. Is the manuscript technically sound, and do the data support the conclusions?

Reviewer #1: Yes

Reviewer #2: Yes

Reviewer #3: Partly

2. Has the statistical analysis been performed appropriately and rigorously?

Reviewer #1: Yes

Reviewer #2: I Don't Know

Reviewer #3: Yes

3. Have the authors made all data underlying the findings in their manuscript fully available?

Reviewer #1: Yes

Reviewer #2: Yes

Reviewer #3: Yes

4. Is the manuscript presented in an intelligible fashion and written in standard English?

Reviewer #1: Yes

Reviewer #2: Yes

Reviewer #3: Yes

Reviewer #1: The role of SPP1 in evaluating the prognosis, immune infiltration, and drug sensitivity of hepatocellular carcinoma

Overall Assessment

This manuscript investigates the role of secretory phosphoprotein 1 (SPP1) in hepatocellular carcinoma (HCC) through a combination of integrated bioinformatics analyses and in vitro validation experiments. Utilizing publicly accessible datasets such as TCGA, GEO, TIMER, HPA, and GDSC, the authors conduct a comprehensive evaluation of SPP1 expression, its prognostic significance, associations with immune infiltration, and potential drug sensitivity, which are further validated through functional assays in HCC cell lines. The study is systematically organized and addresses a critical clinical challenge: the identification of prognostic biomarkers and therapeutic targets in HCC. The research topic is both relevant and potentially transformative. However, several methodological and interpretative issues need to be addressed to enhance the manuscript's rigor.

Major Strengths:

Thorough multi-database validation: The authors adeptly confirm the overexpression of SPP1 by examining data from TCGA alongside six independent GEO datasets, thereby bolstering the robustness and reproducibility of their results.

Comprehensive prognostic modeling: The study utilizes survival analyses, Cox regression, ROC curves, and nomogram construction to create a detailed framework for prognostic assessment.

Analysis of the immune microenvironment: By integrating CIBERSORT, TIMER, ESTIMATE, and immune checkpoint correlation analysis, the research offers mechanistic insights into the interplay between SPP1 and the tumor immune microenvironment.

Functional validation: In vitro experiments, including knockdown and overexpression (CCK-8, colony formation, migration, invasion, and apoptosis assays), support the bioinformatics findings, thereby strengthening the biological plausibility of the study's conclusions.

Comments:

The research was confined to HepG2 cells. To improve the study's strength and reliability, it is advisable to incorporate a broader spectrum of hepatocellular carcinoma (HCC) cell lines. Furthermore, expanding the investigation to include in vivo models would be advantageous.

The mechanistic depth is insufficient; although enrichment analyses suggest the involvement of PI3K-Akt and immune-related pathways, no direct mechanistic experiments, such as pathway inhibition or Western blotting for downstream signaling proteins, were conducted to confirm pathway activation.

The interpretation of immune infiltration is speculative, as conclusions regarding macrophage polarization and CD8+ T-cell suppression lack direct experimental validation through methods such as flow cytometry or immunohistochemistry.

Statistical clarity is required; the DEG threshold is reported as “logFC < 1,” which appears to be a typographical error and should be |logFC| > 1.

Several typographical and spacing errors are present throughout the manuscript and should be corrected for clarity.

Inconsistencies are noted in some figure legends, such as EXOSC10 being mentioned in Figure 1C instead of SPP1.

The Discussion section could be more concise and better focused on the study’s novel contributions.

Ethical approval and cell line authentication statements are not clearly described.

Conclusion: This study highlights the pronounced overexpression of SPP1 in hepatocellular carcinoma (HCC), associating it with unfavorable prognosis, immune modulation, and unique drug sensitivity profiles. The integration of bioinformatics with experimental validation in this research is praiseworthy. However, to further substantiate these findings, additional mechanistic studies, validation across diverse cell lines, and experimental drug sensitivity assays are crucial. Addressing these areas and improving the manuscript's clarity could greatly enhance contributions to HCC research.

Recommendation: Minor revision.

Reviewer #2: Comments to Author: In this manuscript, Cui et al. investigate the potential role of SSP1 in hepatocellular carcinoma using bioinformatics pipelines and cellular assays. The authors have incorporated the necessary controls and appropriately cited the literature; however, it requires some editing, particularly in the placement of commas between reference numbers. Overall, the manuscript is well-structured but would benefit from minor revisions focusing on usage of scientific terminology to enhance the presentation of the findings, especially in the Discussion section.

Comments:

Introduction: Referencing needs editing there should be comma in between 23 same goes with 56,911,1213 throughout the introduction written part.

“In 2020, hepatocellular carcinoma (HCC) had a global population of 910,000 cases and 830,000 deaths, which is a common malignant tumor.1 In terms of global malignant tumors, the number of new cases and deaths ranks sixth and fourth, respectively.1” These two sentences can be merged into one effective sentence.

“Initially, the mRNA expression level of SPP1 was assessed using publicly available databases,……………………………..”. After this author have mentioned that “We also investigated the association between SPP1 expression and immune………….”. This paragraph can be reframed meaning since authors are trying to say that what they have done in this research work.

Something like this “In this study, we initially assessed the mRNA expression level of SPP1 using publicly available databases. Additionally, we investigated the association between SPP1 expression and immune responses”. This approach enables us to establish a clear flow of our research objectives and findings.

In vitro should be in italics.

Materials and Methods: This section is well-written and shows great attention to detail. However, authors have not provided information regarding the expression of genes in hepatocellular carcinoma (HCC) cell lines; rather, they have solely addressed the cultivation of these cells.

Figure legends could be little bit descriptive for example Fig 1, there is no description in the legends about 1D-I. Moreover, for Fig1J authors can mentioned instead of “HPA database, the expression of SPP1 protein is elevated in HCC tissues compared to non-HCC tissues” as shown in left circle or right circle. It is not quite clear for the reader.

Regarding Figures 5 and 6, if the authors are presenting IC50 values, it would be more effective to display them in a table format, as the graphical representation does not add significant value. Alternatively, they could create a representative graph that differentiates between low and high IC50 values and then compile the data in a table, categorizing the drugs into low and high IC50 groups.

In Fig 7C is it SNU-387 or 388 because WB showed as 388.

“The KEGG enrichment pathway primarily included neuroactive ligand-receptor interaction,

cytoskeleton in muscle cells, cytokine-cytokine receptor interaction, and the PI3K-Akt signaling pathway, among others (Figure 3D).” This statement is not very clear in terms of figure representation.

“Results revealed that high SPP1 expression was correlated with significantly higher immune and ESTIMATE scores compared to the low-expression group (Figure 4A).” Which results authors are trying to refer it in here?

It can be re-written as “To validate the findings from the in-silico gene analysis, we examined the expression of SPP1 in HCC cell lines, either through transient transfection or stable integration”.

However, the authors did not specify which method was used for the expression of SPP1 in HCC cell lines, either in the main text or in the figure legends. This information is crucial, as the relevance of the data obtained depends significantly on whether SPP1 expression was transient or stably integrated.

This can be introduced into the introduction “SPP1, located on chromosome 4q13 and consisting of seven exons and six introns, encodes a 294-amino-acid secreted phosphoprotein. In humans, three domains of SPP1 bind to integrins: the RGD domain, the SVVYGLR domain, and the ELVTDFPTDLPAT domain. Among them, the most well-known ones are the RGD domain and the SVVYGLR domain. SPP1 promotes tumor development and progression by binding to CD44 and integrin receptors.” Because these findings regarding SPP1 do not contribute meaningfully to the discussion, as the authors have not conducted any biochemical or biophysical assays to emphasize the significance of these domains.

“In this work, we employed bioinformatics techniques to assess the function of SPP1 in relation to the expression, prognosis, and therapeutic sensitivity of HCC patients, and we verified our findings through specific experiments.” Authors needs to mention what specific experiments.

“Meanwhile, we verified that SPP1 may function as a stand-alone risk factor for assessing the prognosis of HCC patients and is connected to the histological grade, pathological grade, and T stage of tumors.” Authors should rewrite this like In addition to above mentioned findings, we have verified……….

“Consequently, we investigated whether SPP1 is linked to tumor immunity in HCC.” This can be expressed in a more scientific manner, emphasizing author’s unique contributions rationale behind it rather than simply paralleling previous experiments conducted by other groups.

I appreciated the authors' honesty in expressing this “This article still has many of restrictions, though. To begin with, in terms of experiments, all we did was carry out a basic confirmation of SPP1's involvement in HCC. More research is required to determine how SPP1 influences the onset and progression of HCC. Second, even though SPP1 and tumor immune infiltration were shown to be significantly correlated in our study, the exact mechanism behind this finding is still unknown and needs to be confirmed by more research.” However, authors can write this in a more scientific manner. For example “This article, however, presents several limitations. Firstly, our experimental approach was limited to a basic validation of the involvement of SPP1 in hepatocellular carcinoma (HCC). Further investigation is necessary to elucidate the role of SPP1 in the initiation and progression of HCC. Secondly, while our study identified a significant correlation between SPP1 and tumor immune infiltration, the precise mechanisms underlying this relationship remain unclear and warrant additional research for confirmation”.

Reviewer #3: This manuscript investigates the prognostic and immunological significance of SPP1 in hepatocellular carcinoma (HCC) through integrated bioinformatics analyses and limited in vitro validation. The topic is timely and relevant, and the integration of multiple datasets is commendable. However, several points require attention:

Figure Legend: Figure 1 legend incorrectly refers to EXOSC10 instead of SPP1, indicating a likely typographical or copy-editing error that should be corrected.

Overstated Claims: Statements such as “patients with high SPP1 expression are more vulnerable to these medications” are overstated. A more accurate phrasing would be: “predicted drug sensitivity based on transcriptomic modeling.”

Results Section – Potential Pathway of SPP1 Regulation in HCC: The sentence describing cellular component enrichment appears to be duplicated. Specifically, the statement:

“In the cellular component, DEGs were mainly concentrated in the collagen-containing extracellular matrix, the apical part of the cell, the apical plasma membrane, and the synaptic membrane, among others (Figure 3B and 3C)”

is repeated. The authors should remove the duplicate to improve clarity and avoid redundancy.

Tumor Immune Microenvironment: The manuscript addresses the tumor immune microenvironment and immune cell infiltration associated with SPP1 expression. However, the analysis primarily relies on computational estimations. It would strengthen the study to clarify whether any crosstalk between stromal cells, resident liver cells, and infiltrating immune cells has been investigated or demonstrated. In particular, discussion or experimental evaluation of interactions among these cellular components within the tumor microenvironment would be valuable. Furthermore, it is unclear whether any co-culture experiments (e.g., tumor cells with immune or stromal cells) were performed to validate the predicted immune interactions. Including such validation or addressing this limitation would enhance the interpretation of the findings.

EMT and Metastasis: While the manuscript demonstrates that SPP1 promotes HCC cell proliferation, migration, and invasion, there is no discussion of epithelial-mesenchymal transition (EMT) or metastasis, which are key processes in tumor progression. Including analyses or discussion linking SPP1 to EMT markers and metastatic potential, and referencing any available in vivo or clinical data supporting a role for SPP1 in HCC metastasis, would significantly strengthen the manuscript.

.

Reviewer #1: **Yes:**RAMYA SRI SURARAMYA SRI SURARAMYA SRI SURARAMYA SRI SURA

Reviewer #2: No

Reviewer #3: **Yes:**Hamda SiddiquiHamda SiddiquiHamda SiddiquiHamda Siddiqui

---

## [Author Response · Author response to Decision Letter 1]

4 Apr 2026

Journal Requirements

Response: I have made further revisions according to the style of the journal, as detailed in my resubmitted manuscript. If there are any incorrect modifications, please raise them and I will continue to make them carefully.

2. Thank you for stating the following financial disclosure: “the Science and Technology Program of the Joint Fund of Scientific Research for the Public Hospitals of Inner Mongolia Academy of Medical Sciences (Grant No. 2024GLLH0862).” Please state what role the funders took in the study. If the funders had no role, please state: "The funders had no role in study design, data collection and analysis, decision to publish, or preparation of the manuscript." If this statement is not correct you must amend it as needed. Please include this amended Role of Funder statement in your cover letter; we will change the online submission form on your behalf.

Response: I have made revisions to the relevant suggestions regarding the second and third points, as detailed in my resubmitted cover letter. In addition, I have removed the relevant content from the resubmitted manuscript.

Response: I have added the sentence 'All data is in the manuscript and/or supporting information files' to the' Data Availability Statement '.

5. PLOS ONE now requires that authors provide the original uncropped and unadjusted images underlying all blot or gel results reported in a submission’s figures or Supporting Information files. In your cover letter, please note whether your blot/gel image data are in Supporting Information or posted at a public data repository, provide the repository URL if relevant, and provide specific details as to which raw blot/gel images.

Response: I have placed my blot image data in the supporting information and agree to publish the above data.

6. Please include captions for your Supporting Information files at the end of your manuscript, and update any in-text citations to match accordingly.

Response: I have made revisions to it, please refer to my resubmitted manuscript for details.

Response: The reviewer did not make any relevant requests.

Response: I have reviewed the paper I cited again and there are no references that have been retracted.

Reviewer 1:

1. The research was confined to HepG2 cells. To improve the study's strength and reliability, it is advisable to incorporate a broader spectrum of hepatocellular carcinoma (HCC) cell lines. Furthermore, expanding the investigation to include in vivo models would be advantageous.

Response: You are absolutely right, this study is indeed limited to HepG2 cells. The application of more hepatocellular carcinoma cell lines will be more convincing. However, despite funding support, this study has limited resources and can only use HepG2 cells for experiments. If there is an opportunity in the future, more cell lines will be used for research to enhance persuasiveness.

2. The mechanistic depth is insufficient; although enrichment analyses suggest the involvement of PI3K-Akt and immune-related pathways, no direct mechanistic experiments, such as pathway inhibition or Western blotting for downstream signaling proteins, were conducted to confirm pathway activation.

Response: Thank you for your careful review and valuable feedback on this article. The issues you pointed out, such as insufficient mechanism depth and the need for direct mechanism experimental verification, are very relevant and we fully agree with them. But equally, our funds are very limited, and we will further research in this direction once we have sufficient funds in the future.

3. The interpretation of immune infiltration is speculative, as conclusions regarding macrophage polarization and CD8+ T-cell suppression lack direct experimental validation through methods such as flow cytometry or immunohistochemistry.

Response: What you said is correct, I have made modifications based on your suggestions. If there is anything else that you are not satisfied with, please raise it and I will continue to make serious revisions.

Page 15:

Before revise: Consequently, we investigated whether SPP1 is linked to tumor immunity in HCC.

After revise: Therefore, we investigated the relationship between SPP1 expression and immune-related features in HCC.

Page 15:

Before revise: According to our research, SPP1 serves a crucial role in the infiltration of neutrophils, macrophages, myeloid dendritic cells, and B cells.

After revise: In our analyses, SPP1 expression was associated with transcriptional signatures related to neutrophils, macrophages, myeloid dendritic cells, and B cells.

Page 15:

Before revise: Macrophages, which originate from monocytes, play a crucial role in both specific and non-specific immunity, enabling the host to maintain the internal balance of tissues [34, 35]. The phenotypic diversity of macrophages results in variations in their functions when they encounter tumors, and even opposite effects can occur.

Before revise: Macrophages, derived from monocytes, are key effectors of innate and adaptive immunity and exhibit phenotypic plasticity in tumors; different macrophage phenotypes can exert distinct and sometimes opposing effects on tumor biology [34, 35].

4. Statistical clarity is required; the DEG threshold is reported as “logFC < 1,” which appears to be a typographical error and should be |logFC| > 1.

Response: Indeed, this was my mistake. I have already made modifications to it.

Page 6:

Before revise: Two groups of differentially expressed genes (DEGs) were screened using the threshold criteria of logFC < 1, and FDR < .05.

After revise: Two groups of differentially expressed genes (DEGs) were screened using the threshold criteria of |logFC| > 1 and FDR < .05.

5. Several typographical and spacing errors are present throughout the manuscript and should be corrected for clarity.

Response: Thank you for your reminder. The editor of the journal has asked me to further modify the format according to the journal's requirements.

6. Inconsistencies are noted in some figure legends, such as EXOSC10 being mentioned in Figure 1C instead of SPP1.

Response: This is indeed my mistake. I have already made modifications to it.

Before revise: (C) EXOSC10 was significantly elevated in 374 HCC samples compared to the corresponding adjacent liver tissue. In databases GSE45436, GSE54236, GSE121248, GSE76427, GSE64041, and GSE60502, SPP1 expression is higher in HCC samples compared to normal samples.

After revise: (C) SPP1 was significantly elevated in 374 HCC samples compared to the corresponding adjacent liver tissue. In databases GSE45436, GSE54236, GSE121248, GSE76427, GSE64041, and GSE60502, SPP1 expression is higher in HCC samples compared to normal samples.

7. The Discussion section could be more concise and better focused on the study’s novel contributions.

Response: I have made modifications based on your feedback. If further modifications are needed, please let me know and I will continue to make them.

Page 17

Before revise: This study used comprehensive bioinformatics to investigate the connection between SPP1 and HCC. While chemotherapy, targeted therapy, and immune checkpoint inhibitors represent important treatment options for advanced HCC, not all patients respond effectively. Therefore, identifying treatment targets helps to select the most suitable drugs, thereby better implementing precision treatment and enabling patients to achieve higher survival. As a result, SPP1 may be targeted for HCC as a result of this study's research, offering a novel approach to the treatment of HCC patients. This article still has many of restrictions, though. To begin with, in terms of experiments, all we did was carry out a basic confirmation of SPP1's involvement in HCC. More research is required to determine how SPP1 influences the onset and progression of HCC. Second, even though SPP1 and tumor immune infiltration were shown to be significantly correlated in our study, the exact mechanism behind this finding is still unknown and needs to be confirmed by more research.

After revise: In this study, we provide a multi-layered characterization of SPP1 in hepatocellular carcinoma (HCC). Across TCGA, six GEO datasets, and HPA protein data, SPP1 was consistently upregulated at the mRNA and protein levels. High SPP1 expression was associated with poorer overall and progression‑free survival and with more advanced clinicopathologic features. In vitro, SPP1 knockdown reduced HCC cell proliferation, migration, and invasion and increased apoptosis, while SPP1 overexpression had opposite effects.

Page 14

Before revise: SPP1, located on chromosome 4q13 and consisting of seven exons and six introns, encodes a 294-amino-acid secreted phosphoprotein [27]. In humans, three domains of SPP1 bind to integrins: the RGD domain, the SVVYGLR domain, and the ELVTDFPTDLPAT domain [28]. Among them, the most well-known ones are the RGD domain and the SVVYGLR domain. SPP1 promotes tumor development and progression by binding to CD44 and integrin receptors. Through interaction with CD44 and αvβ3 integrin, SPP1 activates the PI3K/Akt/HIF‑1α and PI3K/Akt/NF‑κB signaling pathways, thereby enhancing tumor angiogenesis and epithelial‑mesenchymal transition (EMT), respectively [29,30]. Additionally, SPP1 binding to αvβ3 integrin triggers the JNK pathway, leading to upregulation of MMP2 and MMP9 expression and facilitating tumor invasion [30]. Furthermore, SPP1 attaches itself to α9β1 integrin, initiating the p38 and ERK signaling pathways and increasing the production of COX-2, a protein that promotes the migration of tumor cells [31]. Although SPP1 has been linked to multiple malignancies [32], its prognostic significance and underlying mechanisms in HCC remain incompletely understood.

After revise: SPP1 encodes a 294–amino-acid secreted phosphoprotein located on chromosome 4q13 [27]. SPP1 contains integrin-binding domains including RGD and SVVYGLR [28]. The RGD and SVVYGLR domains are the best characterized. SPP1 promotes tumor progression via interactions with CD44 and integrins. Through interaction with CD44 and αvβ3 integrin, SPP1 activates the PI3K/Akt/HIF‑1α and PI3K/Akt/NF‑κB signaling pathways, thereby enhancing tumor angiogenesis and epithelial‑mesenchymal transition (EMT), respectively [29,30]. Additionally, SPP1 binding to αvβ3 integrin triggers the JNK pathway, leading to upregulation of MMP2 and MMP9 expression and facilitating tumor invasion [30]. Interaction with α9β1 activates p38/ERK and upregulates COX 2, enhancing tumor cell migration [31]. Despite associations with various cancers, SPP1’s prognostic role and mechanisms in HCC are not fully defined [32].

Page 14

Before revise: We also used six independent GEO databases (GSE45436, GSE54236, GSE121248, GSE76427, GSE64041, and GSE60502) for validation. Furthermore, in HCC patients, higher SPP1 expression is associated with worse OS and PFS. Meanwhile, we verified that SPP1 may function as a stand-alone risk factor for assessing the prognosis of HCC patients and is connected to the histological grade, pathological grade, and T stage of tumors.

After revise: We validated SPP1 upregulation using six independent GEO cohorts. Clinically, high SPP1 correlates with poorer OS and PFS. Multivariate analysis indicates SPP1 is an independent prognostic factor and associates with higher histologic grade and T stage.

8. Ethical approval and cell line authentication statements are not clearly described.

Response: Thank you for your reminder, I did indeed miss this point. I have added this section to the manuscript.

Page 5: The data used in this study were all from public databases, and the cell lines used were commercially purchased. Therefore, this study did not require specific ethical approval.

Reviewer 2:

1. Introduction: Referencing needs editing there should be comma in between 23 same goes with 56,911,1213 throughout the introduction written part.

Response: I do have some shortcomings in terms of references, and I have revised them accordingly.

2. “In 2020, hepatocellular carcinoma (HCC) had a global population of 910,000 cases and 830,000 deaths, which is a common malignant tumor.1 In terms of global malignant tumors, the number of new cases and deaths ranks sixth and fourth, respectively.1” These two sentences can be merged into one effective sentence.

Response: These two sentences are indeed a bit redundant, I have merged them together.

Page 3

Before revise: In 2020, hepatocellular carcinoma (HCC) had a global population of 910,000 cases and 830,000 deaths, which is a common malignant tumor. In terms of global malignant tumors, the number of new cases and deaths ranks sixth and fourth, respectively.

After revise: In 2020, hepatocellular carcinoma (HCC), a common malignant tumor, had a global burden of approximately 910,000 new cases and 830,000 deaths, ranking sixth and fourth, respectively, in terms of incidence and mortality worldwide.

3.“Initially, the mRNA expression level of SPP1 was assessed using publicly available databases,……………………………..”. After this author have mentioned that “We also investigated the association between SPP1 expression and immune………….”. This paragraph can be reframed meaning since authors are trying to say that what they have done in this research work.

Response: Thank you for your reminder, I have made further modifications.

Page 4

Before revise: Initially, the mRNA expression level of SPP1 was assessed using publicly available databases, including The Cancer Genome Atlas (TCGA), Gene Expression Omnibus (GEO), and the Human Protein Atlas (HPA). Subsequently, the prognostic value of SPP1 in HCC was further validated with TCGA survival data. We also investigated the association between SPP1 expression and immune cell infiltration within the tumor microenvironment. In addition, the potential of SPP1 as a biomarker for predicting drug sensitivity in HCC was systematically evaluated. To experimentally verify its role, SPP1 was knocked out in HCC cell lines, and its expression was examined in vitro. Collectively, this study elucidates the important functions of SPP1 in HCC progression and therapy.

After revise: This study aims to comprehensively elucidate the role of SPP1 in HCC progression and therapy. To achieve this, we performed a multi-faceted study. First, we assessed the mRNA and protein expression levels of SPP1 in HCC using publicly available databases, including The Cancer Genome Atlas (TCGA), Gene Expression Omnibus (GEO), and the Human Protein Atlas (HPA). Subsequently, the prognostic value of SPP1 in HCC was further validated with TCGA survival d

---

## [Editor Report · Decision Letter 1]

8 Apr 2026

The role of SPP1 in evaluating the prognosis, immune infiltration, and drug sensitivity of hepatocellular carcinoma

PONE-D-26-03810R1

Dear Dr. Long,

We’re pleased to inform you that your manuscript has been judged scientifically suitable for publication and will be formally accepted for publication once it meets all outstanding technical requirements.

Kind regards,

Vinay Kumar, Ph.D.

Academic Editor

PLOS One
---

## [Editor Report · Acceptance letter]

PONE-D-26-03810R1

PLOS One

Dear Dr. Long,

I'm pleased to inform you that your manuscript has been deemed suitable for publication in PLOS One. Congratulations! Your manuscript is now being handed over to our production team.

Kind regards,

on behalf of

Dr. Vinay Kumar

Academic Editor

PLOS One